# Utilizing Genome-Wide mRNA Profiling to Identify the Cytotoxic Chemotherapeutic Mechanism of Triazoloacridone C-1305 as Direct Microtubule Stabilization

**DOI:** 10.3390/cancers12040864

**Published:** 2020-04-02

**Authors:** Jarosław Króliczewski, Sylwia Bartoszewska, Magdalena Dudkowska, Dorota Janiszewska, Agnieszka Biernatowska, David K. Crossman, Karol Krzymiński, Małgorzata Wysocka, Anna Romanowska, Maciej Baginski, Michal Markuszewski, Renata J. Ochocka, James F. Collawn, Aleksander F. Sikorski, Ewa Sikora, Rafal Bartoszewski

**Affiliations:** 1Department of Biology and Pharmaceutical Botany, Medical University of Gdansk, 80-416 Gdansk, Poland; jakrol@windowslive.com (J.K.); renata@gumed.edu.pl (R.J.O.); 2Department of Inorganic Chemistry, Medical University of Gdansk, 80-416 Gdansk, Poland; sylwiabart@gumed.edu.pl; 3Laboratory of the Molecular Bases of Ageing, Nencki Institute of Experimental Biology of the Polish Academy of Sciences, 02-093 Warsaw, Polandd.janiszewska@nencki.edu.pl (D.J.); e.sikora@nencki.edu.pl (E.S.); 4Department of Cytobiochemistry, Faculty of Biotechnology, University of Wroclaw, 50-383 Wroclaw Poland; agi.biernatowska@gmail.com; 5Department of Genetics, UAB Genomics Core Facility, University of Alabama at Birmingham, Birmingham, AL 35233, USA; dkcrossm@uab.edu; 6Faculty of Chemistry, University of Gdansk, 80-308 Gdansk, Poland; karol.krzyminski@ug.edu.pl (K.K.); malgorzata.wysocka@ug.edu.pl (M.W.); anna.romanowska@phdstud.ug.edu.pl (A.R.); 7Department of Pharmaceutical Technology and Biochemistry, Faculty of Chemistry, Gdansk University of Technology, 80-233 Gdansk, Poland; maciej.baginski@pg.edu.pl; 8Department of Biopharmacy and Pharmacodynamics, Medical University of Gdansk, 80-416 Gdansk, Poland; michal.markuszewski@gumed.edu.pl; 9Department of Cell, Developmental and Integrative Biology, University of Alabama at Birmingham, Birmingham, AL 35294, USA; jcollawn@uab.edu; 10Research and Development Centre, Regional Specialist Hospital, 51-154, Wroclaw, Poland; afsikorski@gmail.com

**Keywords:** microtubule, RNASeq, tubulin, cell cycle arrest

## Abstract

Rational drug design and in vitro pharmacology profiling constitute the gold standard in drug development pipelines. Problems arise, however, because this process is often difficult due to limited information regarding the complete identification of a molecule’s biological activities. The increasing affordability of genome-wide next-generation technologies now provides an excellent opportunity to understand a compound’s diverse effects on gene regulation. Here, we used an unbiased approach in lung and colon cancer cell lines to identify the early transcriptomic signatures of C-1305 cytotoxicity that highlight the novel pathways responsible for its biological activity. Our results demonstrate that C-1305 promotes direct microtubule stabilization as a part of its mechanism of action that leads to apoptosis. Furthermore, we show that C-1305 promotes G2 cell cycle arrest by modulating gene expression. The results indicate that C-1305 is the first microtubule stabilizing agent that also is a topoisomerase II inhibitor. This study provides a novel approach and methodology for delineating the antitumor mechanisms of other putative anticancer drug candidates.

## 1. Introduction

The development of cytotoxic chemotherapy into molecularly targeted cancer drug discovery provided an opportunity to deliver successful therapies that could change the lives of a large number of cancer patients. However, reducing the safety concerns of any drug therapy requires a thorough evaluation of the molecular mechanisms of the candidate molecules and, importantly, the prediction of their off-target effects [1,2]. Nevertheless, despite their obvious benefits, early in-vitro pharmacological profiling approaches that address the molecular mechanisms that define molecular anticancer effects are rarely utilized [1]. 

Studies that focus on presenting the molecular mechanisms governing cytotoxicity, which are currently fueling the anticancer development pipeline, often focus on single-pathway models that are difficult to translate into the standard drug development process [1,2]. However, recent developments and the decreasing costs of high-throughput genotyping technologies like deep sequencing [3,4] now allow investigators to address these research limitations. 

Several lines of evidence have indicated that the triazoloacridinone derivative, C-1305 (Figure 1) altered gene and protein expression and promoted apoptosis [5]. Furthermore, the antitumor activity of C-1305 [6,7] was connected with its ability to intercalate to DNA and/or inhibit topoisomerase II activity in in-vitro assays [8]. The genome-wide impact of C-1305, however, and its exact mechanism of action, remain unknown. 

In this study, we used an unbiased approach in lung cancer and colon cell lines (A549 and HTC 116, respectively) to identify universal early transcriptomic signatures of C-1305 cytotoxicity and to highlight the novel pathways responsible for its biological activity. The cancer cell lines chosen for our studies were previously reported as very sensitive to the potent anticancer activity of C-1305, [7,8,9].

Our results identified C-1305-mediated direct microtubule stabilization as the central mediator that leads to apoptosis. Furthermore, we demonstrated that C-1305 promotes G2 cell cycle arrest by modulating gene expression. The results indicate that C-1305 is the first identified microtubule stabilizing agent coupled with topoisomerase II inhibitory activity. Our work provides a framework to understand the effects of C-1305 on cancer cells and serves as a resource for delineating the antitumor mechanisms of other putative anticancer drug candidates.

## 2. Results

### 2.1. Evaluation of C-1305 Cytotoxicity Profile

The synthesis of compound C-1305 was performed by adopting and optimizing literature descriptions [6,10]. C-1305 compound identity and purity were verified as described in the methods section. Next, to determine when the C-1305 began to induce apoptosis, we performed a time-course study and monitored A549 and HTC 116 viability using impedance-based real-time analysis (Figure 2A–E).

The applied real-time monitoring system allows the evaluation of C-1305 dose-dependent changes in a cellular index that represent cells’ viability, attachment, and morphology [11,12,13,14]. The cancer cell lines chosen for our studies were previously used in studies testing the potent anticancer activity of C-1305 [7,8,9].

C-1305 at concentrations below 10 µM had no significant effect on A549 cell growth and survival until approximately 24 h, when the cell index numbers decreased. Furthermore, the C-1305 IC_50_ value calculated based on this real-time assay after 48 h treatment was about 3 µM in A549 (Figure 2A,B). Similar, real-time survival profiles were obtained for C-1305 treatment in HTC 116 cells. C-1305 at concentrations below 10 µM had no significant effect on HTC 116 cell growth and survival until approximately 18 h, when the cell index numbers decreased in a concentration-dependent manner. Furthermore, the C-1305 IC_50_ value, calculated based on this real-time assay after 48 h treatment, was about 10µM in HTC 116 cells (Figure 2C,D). Notably, the C-1305 IC_50_ values obtained with the real-time system corresponded well to the values obtained by the independent MTT assays: 3.08 µM for A549 (after 24 h, *p* = 0.0024) and 9.27 µM for HTC 116 (after 24 h, *p* = 0.0019). Finally, C-1305 effects on cell survival were also tested in immortalized normal human lung epithelial cell line (16HBE14o-). The IC_50_ was 23.66 µM (after 24 h the *p* = 0.0012), and as shown in Figure 2E, the C-1305 concentrations below 5 µM had no significant impact on cell growth.

The data obtained with real-time analysis was used to select appropriate doses for subsequent RNAseq and biochemical analysis. Furthermore, the RNA samples prior to RNA-seq analysis were pre-verified for transcriptomic activation of apoptosis related pathways via qPCR (Figure 3). We focus on p53 and DNA damage signaling, since these pathways were previously proposed to be involved in C-1305 cytotoxicity [8,15]. Since activation of apoptosis transcriptomic signaling should precede its phenotypical effects, we focused on the 24 h exposure time points (base on our real-time assays results) when the cytotoxic effects of C-1305 just started to be significant.

The 24 h exposure of the A549 and HTC 116 to C-1305 at their IC_50_ concentrations significantly induced tumor protein p53 (*TP53*) expression and the downstream gene mRNA expression of Bcl-2 binding component 3 (*BBC3*). Meanwhile, there were no significant changes in these gene transcripts’ levels in 16HBE14o- cells exposed to a non-toxic (3 µM) concentration of C-1305 (Figure 3A,B). Similar elevated patterns were also observed for the mRNAs of DNA damage inducible transcript 3 *(DDIT3*) and growth arrest and DNA damage inducible alpha (*GADD45A*) (Figure 3C,D). However, these transcripts were also slightly induced in 16HBE14o- cells.

Furthermore, C-1305 treatment resulted in increased expression of mitochondrial apoptosis markers Bcl-2-associated X apoptosis regulator (*BAX*) and Bcl-2-associated agonist of cell death (*BAD*) in cancer cells, when compared to controls and 16HBE14o- cells (Figure 3E,F). The mRNA levels, however, of the other mitochondrial apoptosis marker, BH3-interacting domain death agonist (*BID*), and the extrinsic apoptosis marker, Fas associated via death domain (*FADD*), remained unaffected in all models (Figure 3G,H). Taken together, these data confirmed the activation of C3105-related apoptosis transcriptomic pathways in the cancer cell model (A549 and HTC 116) and limited or no activation of apoptotic signaling in the control cell model (16HBE14o−).

### 2.2. RNAseq Global Transcriptome Analysis

Since our real-time analysis of cell growth indicated that 24 h exposure to C-1305 (at IC_50_ concentrations) was sufficient to significantly alter A549 and HTC 116 cell growth and to activate apoptosis-related transcriptional signals, we determined the genome-wide transcriptomic alterations in A549 and HTC 116 upon C-1305 treatment. In brief, the A549 and HTC 116 cells were exposed to 3 µM or 10 µM of C-1305, respectively, for 24 h one day after plating. Furthermore, since our data indicated that a 24 h exposure of noncancer cells, epithelial lung cells (16HBE14o−), to 3 µM C-1305 resulted in minimal cellular damage and loss of viability (Figure 2E and Figure 3), we included the treatment of 16HBE14o− as a negative control in our analysis. Finally, since genomic instabilities were reported for A549 cells [16], DMSO vehicle-treated cells (controls) were obtained after 8 and 24 h of cell culture. Total RNA was extracted from the cells after 24 h exposure to C-1305 and 24 h and/or 8 h exposure to vehicle and subjected to RNA sequencing.

As shown in Figure 4A (Data Set S1), after 24 h exposure to C-1305, 3776 and 1769 genes were dysregulated (by at least 2-fold) in A549 cells. Notably, culturing A549 cells for 16 h in the presence of DMSO vehicle affected the expression of 562 genes. Furthermore, despite no significant cytotoxicity in 16HBE14o- cells, the exposure to C-1305 dysregulated the mRNAs of 2405 genes. Hence, in order to focus on universal transcriptome changes related to C-1305 cytotoxicity, we performed a step-by-step signal selection. As shown in Figure 4, we first corrected the A549-C-1305-treated gene set with the A549-vehicle-treated gene set. Next, the A549-C-1305-treated gene set was corrected with the non-toxic 16HBE14o—C-1305-treated gene set and the analysis was narrowed to the most significant expression changes (*p* ≤ 0.05). This approach resulted in the identification of 2040 and 353 genes dysregulated upon C-1305 treatments in A549 and HTC 116 cells, respectively. In the final selection stage, we selected 153 common genes (26 upregulated and 127 downregulated) for which expression was significantly changed in both of these cancer cell lines upon C-1305 exposure (Data Set S1).

Next, for the selected genes, we performed gene ontology analysis using three different software approaches (Gene Analytics [17], Enricher [18] and WebGestalt [19]). As shown in Figure 4B and Table 1, microtubule cytoskeleton organization and tubulin binding were the most significant gene ontology (GO) molecular clusters among the dysregulated genes. This highlighted an important role for the genes involved in chromosome segregation, mitotic spindle formation and cell cycle regulation (Table 1).

Within this cluster of gene transcripts, the following genes’ mRNA levels were downregulated upon C-1305 exposure: Aurora kinase B (*AURKB),* Polo-like kinase 1 *(PLK1),* cyclin-dependent kinase 1 (CDK1), kinesin family members 1C, 11, 14, 15, 18A, 18B, 20A, 22, 23 and 4A *(KIFC1);* cell division cycle-associated 8 *(CDCA8),* baculoviral IAP-repeat-containing 5 *(BIRC5); cyclins B1 and B2 (CCNB1* and *CCNB2),* cell-division cycle protein 20 *(CDC20), Lamin B1 (LMNB1),* centromere proteins F, J, F, K, N, and Q (*CENPF*, *CENPJ, CENPF, CENPK, CENPN and CENPQ*); family with sequence similarity 83 member D (*FAM83D*), growth arrest-specific 2-like 3 (*GAS2L3*), nucleolar and spindle-associated protein 1 (*NUSAP1*), protein regulator of cytokinesis 1 (*PRC1*), spindle and kinetochore-associated complex subunit 1 (SKA1) stathmin 1 (*STMN1*) (Figure 4B). Notably, the upregulated gene set were tumor protein P53-inducible protein 3 (*TP53I3),* tumor Protein P53-inducible nuclear protein 1 *(TP53INP1)* and Snail family transcriptional repressor 2 *(SNAI2)* and these had no significant assignment to the cellular signaling pathways.

### 2.3. C-1305 Affects Microtubule Cytoskeleton Through the Induction of Tubulin Polymerization

Based on the results of the results of the RNAseq-based approach, we speculated that C-1305 cytotoxicity may result from the direct disorganization of the microtubule skeleton. This hypothesis was further supported by the good correlation of C-1305’s transcriptomic profile with the transcriptomic changes reported for other tubulin-directed anticancer drugs, including nocodazole and paclitaxel (Data Set S1). C-1305-related interruptions of the microtubule network could consequently cause defects in mitotic spindle assembly and chromosome segregation leading to the deregulation of cell cycle and eventually cell death. To test this hypothesis, we performed molecular docking of C-1305 to the α/β tubulin dimer crystal structure (Figure 5). As a control, the molecular docking was calculated for paclitaxel (tubulin polymerization stabilizing agent [58]) using the same parameters (Appendix A). As shown in Figure 5, C-1305 could bind at multiple locations to both tubulin dimer α and β subunits, with the free energy comparable to that calculated for paclitaxel (−7.1466 kcal/mol). Notably, the predicted C105 binding sites, despite being located at the luminal side of the tubulin dimer, do not overlap with the paclitaxel binding niche or the colchicine binding site [59,60].

The molecular docking results strongly supported the hypothesis that C-1305 could directly affect tubulin polymerization. To further examine this possibility, we performed an in-vitro tubulin polymerization assay.

As shown in Figure 6A, C-1305 strongly stabilized the tubulin polymerization process in a dose-dependent manner. Paclitaxel and vinblastine were used as a control for stabilizing and destabilizing the polymerization of tubulin, respectively. The calculated effective concentration of C-1305 on tubulin polymerization (EC_50_) was 2.74 µM (Figure 6B). Notably, although C-1305, like paclitaxel, promoted tubulin polymerization and possibly stabilized microtubules, these two compounds affected the polymerization process with different kinetics (Figure 6C). The most significant difference was observed in the time required to achieve exponential microtubule polymerization (Lag), which was 1 minute for paclitaxel and 4–8 min for C-1305. Furthermore, both C-1305 and paclitaxel significantly and comparably increased tubulin polymerization rate as well as higher levels of polymerized tubulin at the equilibrium phase (Figure 6D). Taken together, the data indicate that C-1305 acts as an efficient tubulin polymerization stabilizing agent in vitro.

Notably, the molecular docking studies predicted that the C-1305 does not compete for tubulin’s binding site with paclitaxel, and the kinetics of these two compounds’ impact on tubulin polymerization differed as well. This observation suggests that C-1305 effects on tubulin polymerization could result from different mechanisms of interaction than for paclitaxel. Hence, to further examine this hypothesis, we analyzed the combined effects of C-1305 and paclitaxel co-treatment on tubulin polymerization in vitro. As shown in Figure 6E, these compounds complemented each other’s effects on tubulin polymerization, significantly improving the polymerized tubulin level at equilibrium phase and decreasing the lag time during the co-treatment (Figure 6F). The complementary effects were also well reflected in the maximal velocity of tubulin polymerization, as shown in Appendix A, which was significantly higher than for any of these compounds alone (Figure 6G).

We then tested whether C-1305 had similar effects on tubulin polymerization in A549 cells exposed to this compound for 6 h and 24 h, given that these cells were more sensitive to the compound treatments as shown by lower IC_50_ values when compared to HTC 116. As shown in Figure 7A and Appendix A, C-1305 treatment (equal to or above 1 and 3 µM concentrations for 24 h and 6 h, respectively) dramatically reduced the soluble tubulin fraction and led to a significant accumulation of the polymeric tubulin fraction. Notably, the effects of 5 µM C-1305 were comparable to the 10 µM paclitaxel treatment (Figure 7A).

We also analyzed C-1305’s effects on the microtubule network organization in A549 cells exposed for 24 h to C-1305. Paclitaxel and vinblastine were used as polymerization stabilizing and destabilizing reference controls. As shown in Figure 7B, the C-1305 treatments strongly interrupted the microtubule network. Similar effects of the C-1305 treatments were also observed in the HTC 116 cells (Appendix A). These were both detected with antibodies against the α and β subunits of the tubulin dimer and led to similar results in the paclitaxel treated cells by the thickening of microtubule fibers.

Collectively, the biochemical and immunochemical data confirmed the C-1305-specific impact on tubulin polymerization in A549 cells. To test for the impact of C-1305 treatment on cell cycle progression, the C-1305-treated A549 cells were arrested in the G2 phase of the cell cycle (Figure 8).

To distinguish cells arrested in G2 compared to in M phase, we analyzed the mitotic index using MPM-2 assay. Cells were stained with MPM-2 antibody, which recognized a set of phospho-proteins expressed specifically during mitosis, and then they were measured by flow cytometry. As shown in Figure 8A,B in the A549 cell line treated with C-1305, there was a decrease of MPM-2-positive cells in a time-dependent manner. Meanwhile, after paclitaxel, an accumulation of cells in mitosis was observed. This would suggest that C-1305 caused cell cycle blockade rather in G2 then during mitosis. This observation is in agreement with previous studies reporting C-1305-induced cell arrest at the G2/M checkpoint [15,68]. Notably similar effects on cell cycle arrest were reported for the microtubule network interrupting agents [69,70,71,72].

## 3. Discussion

Rational drug design, together with in-vitro pharmacology profiling, constitute the gold standard of drug development pipelines that allow for the cost-effective reduction of safety-related drug attrition at an early stage during development. However, precise chemical targeting is difficult due to potential adverse side effects and thus requires an understanding of the molecular mechanisms underlying the molecule’s biological activity. Hence, although a large number of potential compounds have been in the drug pipeline, relatively few of them have reached clinical trials or contributed to therapy development. Consequently, many potent promising compounds were left behind or forgotten due to an inability to explain their biological activities in numerous complex gene pathways and cell types. However, the recent development and increasing affordability of genome-wide next-generation technologies provides an opportunity for non-biased determination of the molecular changes brought about by a compound’s multiple activities, and thus revalidation and testing of these forgotten drug candidates is now possible.

In our approach, we exploited label-free, real-time monitoring of cell survival and genome-wide RNA-Seq to revalidate molecular mechanisms underlying the cytotoxicity of triazoloacridone C-1305 in two cancer cell lines. The compound was first synthesized at Gdansk University of Technology in late 1980s [6], during the development of synthetic DNA-interacting agents, and displayed robust cytotoxic and cytostatic effects towards both cancer cell lines and solid tumors in vitro [7]. Despite non-cellular studies that confirmed the ability of C-1305 to covalently bind DNA (preferably in guanidine-rich regions) [73], this phenomenon was not cytotoxicity correlated [74]. In the early 2000s, subsequent research focused on the impact of C-1305 on DNA topoisomerase II activity [8], a crucial regulator of cellular DNA topology [75]. These studies indicated that C-1305 had very mild and “unusual” topoisomerase II poison activity in vitro, but these results did not clarify the observed robust cytotoxic effects [8]. Hence, despite many other, often contradictory, reports [76,77,78], the mechanism of C-1305-induced apoptosis remained undefined.

Since the vast majority of previous attempts to explain C-1305 biological activity have been based on steady-state cytotoxicity assays (often after 72 h exposure), we utilized real-time monitoring of cell survival to determine the time course of C-1305 activity. The results indicate that a significant cell growth inhibition begins at about 24 h after C-1305 administration, which confirms previous studies [8,15]. This was demonstrated in both cancer cell lines and indicated a significant induction of the apoptotic markers’ mRNA levels. Furthermore, comparing the mRNA expression of these markers in cancer cell lines with the mRNA levels in non-cancer cells, we find that their expression correlates well with the observed cytotoxic effects of the C-1305 in the cancer cell lines.

Taking advantage of these differences, we selected appropriate conditions to follow the C-1305-induced changes in the transcriptome with next-generation sequence profiling. Since these RNA-seq results show dramatic differences in C-1305 transcriptomic profiles depending on the cell line tested, we focused on changes in gene expression on mRNAs that were independent of cancer cell line type or vehicle, and thus potentially related to the apoptotic activity of C-1305. This approach resulted in the selection of 153 transcripts that could potentially determine the mechanism of C-1305 activity. The selected mRNAs were mainly related to the microtubule cytoskeleton organization and tubulin binding and critical for the regulation of chromosome segregation, mitotic spindle formation, and cell cycle regulation. Taken together, these data suggest that C-1305-induced apoptosis results from the direct disorganization of its microtubule cytoskeleton and related deregulation of its cell cycle.

The most straightforward explanation of such effects of C-1305 treatment would depend on this compound’s direct interaction with tubulin. The hypothesis that C-1305 could directly affect tubulin polymerization was strongly supported by the molecular-docking-based identification of this compound’s binding locations within α/β tubulin dimers. The in vitro tubulin polymerization assays demonstrated that C-1305 efficiently stabilized the tubulin polymerization process in a dose-dependent manner. Notably, the C-1305-related increase in tubulin polymerization rate, polymerized tubulin level at equilibrium phase, and velocity were comparable to those observed for paclitaxel, a classical tubulin polymerization stabilizing agent. Furthermore, our analysis indicated that C-1305 and paclitaxel do not compete for the same tubulin binding sites, and that these two compounds are additive on tubulin polymerization. This suggests that C-1305 could potentially support paclitaxel-based therapies; however, this hypothesis will require further studies. Finally, we confirmed the proposed tubulin polymerization-related mechanism of C-1305 biological activity using biochemical and immunochemical analyses. The C-1305 treatment resulted in polymeric tubulin accumulation in cells, and strongly interrupted microtubule networks, in a similar manner to paclitaxel, and was accompanied by cell cycle arrest in the G2 phase. Upon C-1305 treatment, there was no accumulation of mitotic marker MPM-2, which was observed in paclitaxel-treated cells. However, it cannot be excluded that the differences between C-1305 and paclitaxel cell effect could be due to the different concentrations of drugs. The observed G2 arrest was not only previously reported for C-1305 [15], but also for many other tubulin destabilizing agents [69,70,71,72]. Support for our results also comes from previous reports that cells depleted of poly(ADP-Ribose) polymerase 1 (*PARP1*) or functional p53 are more sensitive to C-1305 and cell cycle arrest [15]. Notably, both p53 and PARP-1 are crucial regulators of the centrosomes, which play an important role in microtubule function and cell division [79,80,81,82]. Other studies have shown that loss of p53 involves extensive alterations in microtubule composition and dynamics [83], and confers sensitization to paclitaxel [69], similar to the PARP inhibitors [84,85]. Finally, numerous promising anti-cancer compounds were shown to be both topoisomerase poisons and anti-microtubule agents, having a similar pleiotropic activity to C-1305 [86,87,88]. Notably, however, since all these are topoisomerase inhibitors that destabilize microtubules, C-1305 is, to the best of our knowledge, the first known molecule that shares both topoisomerase inhibition and microtubule-stabilizing properties. Nevertheless, further studies are necessary to establish if such an unusual activity of anti-cancer agent can have beneficial effects [88].

## 4. Materials and Methods

### 4.1. Cell lines and Culture Conditions

Human bronchial epithelial 16HBE14o− cells were obtained as previously described (Sigma-Aldrich, catalog no. #SCC150, Poznań, Poland [89]. Human colorectal carcinoma HTC 116 (ATCC CCL-247) cells and human small lung carcinoma A549 cells (ATCC CCL-185) were obtained from ATCC. 16HBE14o− cells were cultured in Minimum Essential Modified Eagle’s Medium (Invitrogen, Warsaw, Poland) with 10% fetal bovine serum (FBS), while HTC 116 cells were cultured in McCoy’s 5a Medium (Invitrogen) with 10% FBS. A549 cells were cultured in RPMI 1640 media supplemented with 10% FBS in a humidified incubator at 37 °C in 5% CO_2_. All experiments were conducted at a final confluence of 70–80%.

### 4.2. Chemicals

5-{[3-(Dimethylamino)propyl]amino}-8-hydroxy-6*H*-[1,2,3]triazolo[4,5,1-*de*]acridin-6-one (C_18_H_19_N_5_O_2_, M_W_ = 337.38 g/mol) (compound C-1305) was synthesized in several steps by adopting an optimization of literature descriptions [6,10]. In the first step 6-chloro-3-nitro-2-{[4-(benzyloxy)-phenyl]amino}benzoic acid was obtained by Ullman condensation of 2,6-dichloro-3-nitrobenzoic acid and 4-(benzyloxy)aniline hydrochloride in the presence of triethylamine in anhydrous ethanol at reflux. The next step was cyclization of the above anthranilic acid derivative by the action of phosphorus oxychloride at reflux. Subsequent stages of the synthesis were removal of the benzyl group from the 8-OH group, reduction of 4-nitroacridin-9(10*H*)-one derivative to the respective 4-aminoacridin-9(10*H*)-9-one, followed by diazotization of the 4-amino group. The last step of the C-1305 synthesis route was the substitution of the halogen atom (5-Cl) in the latter compound by 3-(dimethylamino)propan-1-amine. Purification of C-1305 was performed by applying gravitational liquid chromatography, using silica gel as a stationary phase and gradient type of elution (chloroform/methanol/ammonia mixtures). Detailed information concerning the synthesis of compound C-1305 and its isolation in chemically pure form is provided in Electronic Appendix A (ESI) (Appendix A).

The compound purity and identity were verified with RP-HPLC, elemental analysis, ^1^H NMR spectroscopy and HR MS spectrometry. Representative chromatograms and spectra together with experimental settings are presented in Appendix A (ESI). Summary of C-1305 analyses: RP-HPLC: 98.2% (average area under the main signal, *n* = 3); elemental analysis (% found/calculated): C = 63.76/64.08, H = 5.64/5.68, N = 20.20/20.76; ^1^ H NMR (500 MHz, CDCl_3_, room temperature): δ 1.80–1.85 ppm (quintet, 2H, *J* = 6.8 Hz); 2.18 ppm (s, 6H); 2.36 ppm (t, 2H, *J* = 6.9 Hz); 3.57–3.61 ppm (q, 2H, *J* = 6.4 Hz); 7.16 ppm (d, 1H, *J* = 9.3 Hz); 7.42 ppm (dd, 1H, *J_1_* = 8.8 Hz, *J_2_* = 3 Hz); 7.74 ppm (d, 1H, *J* = 2.4 Hz); 8.29 ppm (d, 1H, J = 9.3 Hz); 8.35 ppm (t, 1H, *J* = 9.3 Hz); 9.46 ppm (t, 1H, *J* = 6.4 Hz, NH); 10.3 ppm (s, 1H, OH). ESI-QTOF MS: m/z = 338.23 (M+1)^+^, 310.22 (M+1−28)^+^. UV-Vis (H_2_O/DMSO = 9/1 v/v): 258.2 nm (ε = 55060 M^−1^ cm^−1^), 298.5 nm (ε = 36580 M^−1^ cm^−1^), 405.1 nm (ε = 43253 M^−1^ cm^−1^), 422.2 nm (ε = 44570 M^−1^ cm^−1^). The C-1305 chemical formula and 3D structure are provided in Figure 1.

The two independent batches of C-1305 were tested. C-1305 was stored in the dark at 4 °C. Prior to the experiments, the compound was dissolved in DMSO at 3 mM stock solution. Paclitaxel was purchased from Cytoskeleton (BK006P, Cytoskeleton Inc., Denver, CO, USA) and vinblastine (catalog no. V1377) and DMSO (catalog no. S-002-D) were purchased from Sigma-Aldrich. Paclitaxel was dissolved in DMSO at 2 mM stock solution and stored at −70 °C. Vinblastine was stored as 100 µM stock in water at −20 °C.

### 4.3. Isolation of RNA and Small ncRNA

Total RNA containing the small ncRNA fraction was isolated using miRNeasy kit (217004, Qiagen, Wroclaw, Poland ). RNA concentrations were calculated based on the absorbance at 260 nm. RNA samples were stored at −70 °C until use.

### 4.4. Next Generation RNA Sequencing Analyses

Cells treated with C-1305 or DMSO vehicle (as specified in the Results section) were used for the RNA isolation and analyses. Following total RNA isolation, samples were validated with quantitative real-time PCR for apoptosis activation prior to further analysis. Following rRNA depletion, the remaining RNA fraction was used for library construction and subjected to 75-bp single-end sequencing on an Illumina HiSeq 2000 instrument (San Diego, CA, USA). Sequencing reads were aligned to the Gencode human reference genome assembly (GRCh38 p7 Release 25) using STAR version 2.5.3b [90]. Transcript assembly and estimation of the relative abundance and tests for differential expression were carried out with HTSeq-count version 0.9.1 and DESeq2 for those samples with biological replicates, and with Cufflinks and Cuffdiff for those samples without biological replicates [91,92,93]. The resulting data were validated with quantitative real-time PCR.

The GeneAnalytics™ web server (geneanalytics.genecards.org) [17] was used to place the Next Generation Sequencing (NGS) results into a physiological context. Furthermore, the analyses were limited to experimentally verified interactions and no extended gene enrichment set analyses were performed. The results of GeneAnalytics analysis were further verified with Enricher [18] and WebGestalt [19]. Furthermore, the transcript changes crucial for GO assignment were independently verified for these gene level changes upon 8 h and 24 h exposure of A549 cells to C-1305, with qPCR.

### 4.5. Measurement of mRNA Levels Using Quantitative Real-Time PCR (qRT-PCR)

We used TaqManOne-Step RT-PCR Master MixReagents (Applied Biosystems, Carlsbad, CA, USA as described previously using the manufacturer’s protocol [11]. The relative expressions were calculated using the comparative relative standard curve method [94]. We used TATA-binding protein (*TBP*) *mRNA,* TFRC and 18S as the relative controls for our studies. TaqMan probes ids used were: *DDIT3*—Hs00358796_g1; *GADD45A*—Hs00169255_m1; *BBC3*—Hs00248075_m1; *TBP*—Hs4332659_m1; *18 S*—Hs99999901_s1; *FADD* Hs04187499_m1; *BID* Hs00609632_m1; *TP3* Hs01034249_m1; *BAX* Hs00180269_m1; *BAD* Hs0018930_m1; *TFRC* Hs00951083_m1.

### 4.6. Cell Viability Assays

For real-time monitoring of cell viability, we used the Roche xCeligence system as we described previously [95]. Briefly, 16HBE14o− cells (2,000 cells per well), A549 cells (7,500 cells per well) or HTC 116 cells (15,000 cells per well) were seeded in 16-well PC plates 12 h prior to the treatment. Control cells were cultured in the presence of DMSO vehicles. Treated cells were incubated with different concentrations of C-1305 for the next 48 h, and every 15 min the cell conductances (cell index) were recorded. All experiments were performed in triplicate with two independent repeats. Control cells were cultured in the presence of DMSO vehicles. RTCA software v. 1.2.1 (ACEA, Biosciences Inc., San Diego, CA, USA) was used to calculate the normalized cell index and the cells’ growth curve slopes.

MTT assays of cell viability were performed in 96-well plates. One day after plating, 16HBE14o− cells (15,000 cells per well), A549 cells (7,500 cells per well) or HTC 116 cells (2,000 cells per well) were cultured with different concentrations of C-1305. After a 24-hour incubation with the specified compounds, medium containing 1 mg/mL MTT was added to cells for a final concentration of 0.5 mg/mL and incubated at 37 °C for 4 h. The medium was aspirated, and the formazan product was solubilized with DMSO. The absorbance at 630 nm (background absorbance) was subtracted from absorbance at 570 nm for each well. There were six replicates for each tested concentration. DMSO was used as control and the concentration in the medium did not exceed 0.1%. The IC_50_ values were calculated from dose–response curves using the four-parameter logistic function.

The permeability of C-1305 to cross the cell membrane was reported previously [9] and also was confirmed with confocal microscopy (Appendix A)

### 4.7. Molecular Docking

In the absence of crystal structures that would indicate an exact location for the C-1305 ligand, a blind docking was performed for C-1305 with the use of the 1TUB PDB file as a receptor [62]. The crystal structure contained one paclitaxel with 420 amino acid residues and GTP in α-tubulin and another with 427 amino acid residues and GDP in β-tubulin. The paclitaxel was removed from the complex before docking. Blind docking was performed with the use of the SwissDock web server [96] based on the docking software EADock DSS [97], whose algorithm consists of the following steps: many binding modes are generated either in a box (local docking) or in the vicinity of all target cavities (blind docking). Simultaneously, their CHARMM energies are estimated on a grid and the binding modes with the most favorable energies are evaluated with fast analytical continuum treatment of solvation (FACTS) and clustered. The mol2 file of C-1305 and colchicine molecules with all hydrogens and 3D coordinates was generated using UCSF Chimera and MarvinSketch. The program Avogadro, version 1.2.0, was used to perform geometric optimization of unbound C-1305 with 500 steps of steepest descent using the MMFF94 force field [98]. The data for tubulin dimers with complexed colchicine molecules was obtained by engineering the crystal structure of colchicine-bound tubulin that was downloaded from the protein data bank (4O2B, PDB)[99], (Appendix A).

### 4.8. In Vitro Tubulin Polymerization Assays

The direct effect of C-1305 on tubulin polymerization was determined by turbidity using a microtubule polymerization assay kit (BK006P, Cytoskeleton Inc., Denver, CO, USA) following the manufacturer’s protocol [100]. Briefly, bovine brain tubulin (3.0 mg/mL) in polymerization assay buffer (80 mM Pipes, pH 6.9, 2 mM MgCl_2_, 0.5 mM EGTA, 1 mM GTP, 10% glycerol) was added to pre-warmed 96-well microtiter plates at 37 °C containing compound at various concentrations in DMSO (2% final). After mixing, the rate of polymerization at 37 °C was followed by absorption at 340 nm for 45 min at 1 min intervals, using a thermostated Tecan Infinite M200Pro microplate spectrophotometer (Tecan Group Ltd., Männedorf, Switzerland). For this assay, paclitaxel was used as the positive control and vinblastine (Sigma-Aldrich) was used as a reference inhibitor. All experiments were repeated three times and the measurements and calculations were conducted with Tecan Magellan V7.0 software, Statistica 13.1 (Dell Inc., 2016, Palo Alto, CA, USA). The half-maximal effective concentration (EC_50_) was defined as the concentration of the drug that gives half-maximal response between the baseline and maximum after a specified exposure time.

### 4.9. Determination of the Monomeric and Polymeric Tubulin with Western Blot

Levels of soluble (monomeric, cytosolic) and polymeric (insoluble, cytoskeletal) tubulin were determined in A549 cells as was described previously [101], with some modification. Briefly, A549 cells at 70% confluency were incubated with or without different concentrations of C-1305 in 6-well plates for 24 h. After 24 h, the cells were washed twice in PBS and then incubated for 4 min at 37 °C in 150 μL of microtubule stabilization buffer (25 mM Hepes, pH 6.9, containing 2 mM MgCl_2_, 2 mM EGTA, 0.5% Triton X-100, 20 μL protease inhibitor mixture (Halt Protease Inhibitor Cocktail (100×), cat. 78430, Thermo Scientific), and 25% glycerol to permeabilize the cells and release the soluble tubulin. The cell lysates were vortexed briefly, and the soluble fractions were separated by centrifugation (15,000× *g* for 10 min at room temperature). The supernatant (200 μL) containing the soluble tubulin was transferred to a new tube. Next, the remaining cytoskeletal fractions (polymeric tubulin) were lysed with ~200 μL of 1× RIPA buffer and protease inhibitor. Soluble and polymerized tubulin samples were stored at −80 °C for future analyses.

The effect of C-1305 on the polymerized amount of tubulin in A549 cells was analyzed by SDS-PAGE (Mini-PROTEAN TGX Stain-Free, Bio-Rad, Warsaw, Poland) and western blot analysis. The protein concentrations for both the fractions of tubulin were measured by DC protein assay kits (Bio-Rad). Protein (20 μg) from both the soluble and polymeric fraction was taken and subjected to electrophoresis. The protein band from gels were transferred onto the poly(vinylidenedifluoride) (PVDF) membrane via the semidry system. After blocking with 5% bovine serum albumin (BSA) in TBST (TBS containing 0.1% Tween 20) for 60 min, the membrane was immunoblotted overnight at 4°C with primary antibodies against β-tubulin (ab18207, Abcam, Cambridge, UK). The membranes were then probed with (H + L) HRP-conjugated secondary antibodies (Cat# 170-6515, Bio-Rad) for 60 min at room temperature. The intensity of the bands was analyzed using ChemiDock system and Image Lab software v. 4.1 (Bio-Rad) and Gel-Pro Analyzer v 4.0 (Media Cybernetics, L.P., Silver Spring, MD, USA).

### 4.10. Immunofluorescence Microscopy

Briefly, A549 cells were grown in 6-well plates (8 × 10^4^ cells/well) and serum-starved for 16 h before treatment with 3 or 10 µM C-1305/paclitaxel/vinblastine respectively for 24 h in a humidified cell culture chamber (37 °C, 5% CO_2_). After incubation, cells were washed twice with PBS and fixed in 2% formaldehyde for 15 min, washed and permeabilized in 0.1% Triton X-100 for 15 min at 4 °C, washed again then blocked in FBS for 30 min. Immunostaining was carried out with appropriate primary antibodies against α-tubulin (catalog no. ab52866, Abcam) or β-tubulin (catalog no. ab18207, Abcam), followed by staining with secondary Alexa-633 antibodies (catalog no. A-21071 Thermofisher Scientific, Warsaw, Poland). Cells were washed as above and incubated for 15 min with mounting medium containing DAPI (ab104139, Abcam). Images were acquired on an LSM 510 META microscope (Carl Zeiss, GmbH, Jena, Germany) using a PLAN-APOCHROMAT 63×/1.4 OIL DIC M27 objective. Image acquisition was performed using ZEN 2009 Light Edition software. Representative cells were chosen to indicate significant differences between controls and treated cells.

### 4.11. Mitotic Index

Mitotic index was estimated by the MPM-2 assay. The anti-phosphorylated Ser/Thr-Pro MPM-2 antibody (Merck Millipore, Poznań, Poland) recognizes a phosphorylated epitope in proteins that are phosphorylated at the onset of mitosis. After treatment with C1305 (IC_50_ dose) or paclitaxel (500 nM) for 6 or 24 h, the medium was discarded, and the cells were collected by trypsinization and fixed in 4% paraformaldehyde (Sigma Aldrich) for 15 min at room temperature and permeabilized with ice-cold 70% ethanol, then kept until staining at −20 °C. For cell staining, the cells were treated with 3% BSA in PBS with 0.1% Triton X-100, and incubated with the anti-MPM-2 antibody (1:1000) then with the secondary antibody, Alexa Fluor 488 goat anti-mouse IgG (Invitrogen, Thermofisher Scientific) (1:1000). The expression of MPM-2 phospho-proteins was determined by flow cytometry using the CellQuest Software; 10,000 events were counted for each sample (FACSCalibur, Becton Dickinson Warsaw, Poland).

### 4.12. Statistical Analysis

Results were expressed as means ± standard deviations (SD). Statistical significance among means was determined using the Student’s *t*-test (two samples, paired and unpaired) after verification of normal distribution of data by Shapiro–Wilk test or the Kruskal–Wallis One Way Analysis of Variance on Ranks [102] and Dunn’s Method [103] (ANOVA on ranks).

### 4.13. Data Accessibility

Deep sequencing data were deposited in Gene Expression Omnibus (GEO) at accession number: GSE143649 [104].

## 5. Conclusions

In summary, although further studies are necessary to test the details of the molecular relationship between C-1305 and the microtubule cytoskeleton and how it can be translated into therapy, the non-biased approach presented here resulted in the identification of a novel mechanism for C-1305-induced apoptosis. The results demonstrate that this triazoloacridone, besides being an unusual topoisomerase poison, binds to tubulin and stabilizes the microtubule network and simultaneously induces cell cycle arrest. Finally, taking into account that in the pre-NGS era, numerous promising and advanced molecules were abandoned due to problems with determining their mechanism of action as well as potential side effects, we provide an example of a modern approach that could result in their reinstatement in the drug development pipeline.

## Figures and Tables

**Figure 1 cancers-12-00864-f001:**
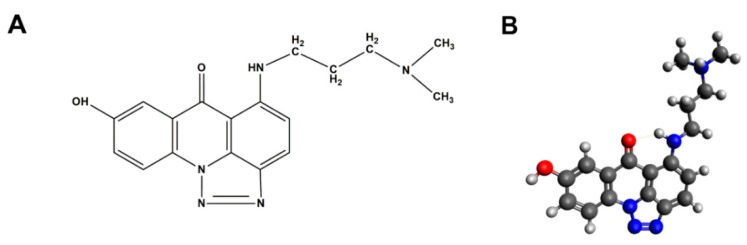
Triazoloacridinone derivative, C-1305. (**A**) Chemical formula of 5-((3-(dimethylamino)propyl)amino)-8-hydroxy-6H-[1,2,3]triazolo[4,5,1-de]acridin-6-one (C_18_H_19_N_5_O_2_)—C-1305; (**B**) Predicted 3-dimensional (3D) structure of C-1305. The 3D structure was generated as described under materials and methods section (red-oxygen, blue-nitrogen, black-carbon, grey-hydrogen).

**Figure 2 cancers-12-00864-f002:**
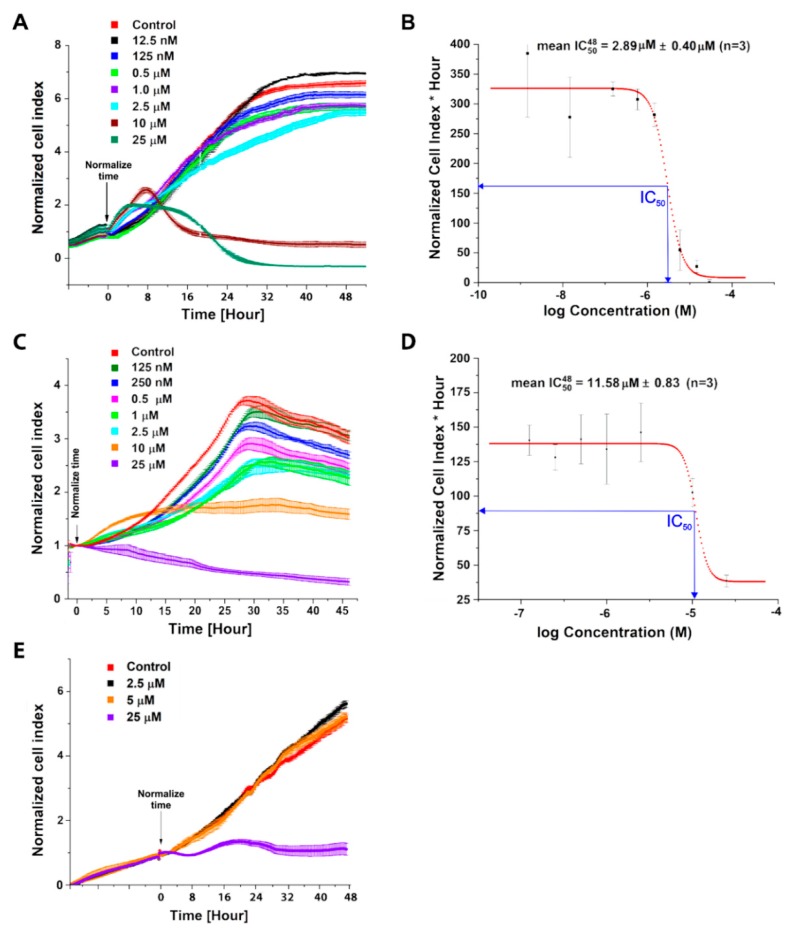
Evaluation of C-1305 cytotoxicity profile. (**A**) Real-time cell analysis of C-1305’s effects on survival of A549 cells. The cell conductances (expressed as normalized cell index) of A549 cells were accessed every 15 min following 48-hour treatment with C-1305 at different concentration 25 µM, 10 µM, 2.5 µM, 1 µM, 0.5 µM, 125 nM, and 12.5 nM. The conductances were normalized to the last value prior to the experiment start. (**B**) IC_50_ curve for compound C-1305 as determined by real-time cell analysis. IC_50_ was calculated after 48 h, as mean and standard deviation (SD), from three independent measurements (*n* = 3). (**C**) Real-time cell analysis of C-1305’s effects on HTC 116 cell survival. The cell conductances (expressed as normalized cell index) of HTC 116 cells were accessed every 15 min following 48-hour treatment with C-1305 at different concentrations: 25 µM, 10 µM, 2.5 µM, 1 µM, 0.5 µM, 250 nM, and 125 nM. The conductances were normalized to the last value prior to the experiment start. (**D**) IC_50_ curve for compound C-1305 as determined by real-time cell analysis. IC_50_ was calculated after 48 h as a mean and standard deviation (SD) from 3 independent measurements (*n* = 3). (**E**) Real-time cell analysis of the effects of increasing concentrations of C-1305 on 16HBE14o− cell survival. The cell conductances (expressed as normalized cell index) of 16HBE14o− cells were accessed every 15 min following 24-hour treatment with C-1305 at different concentrations: 25 µM, 2.5 µM, 5 µM. The conductances were normalized to the last value prior to the experiment start.

**Figure 3 cancers-12-00864-f003:**
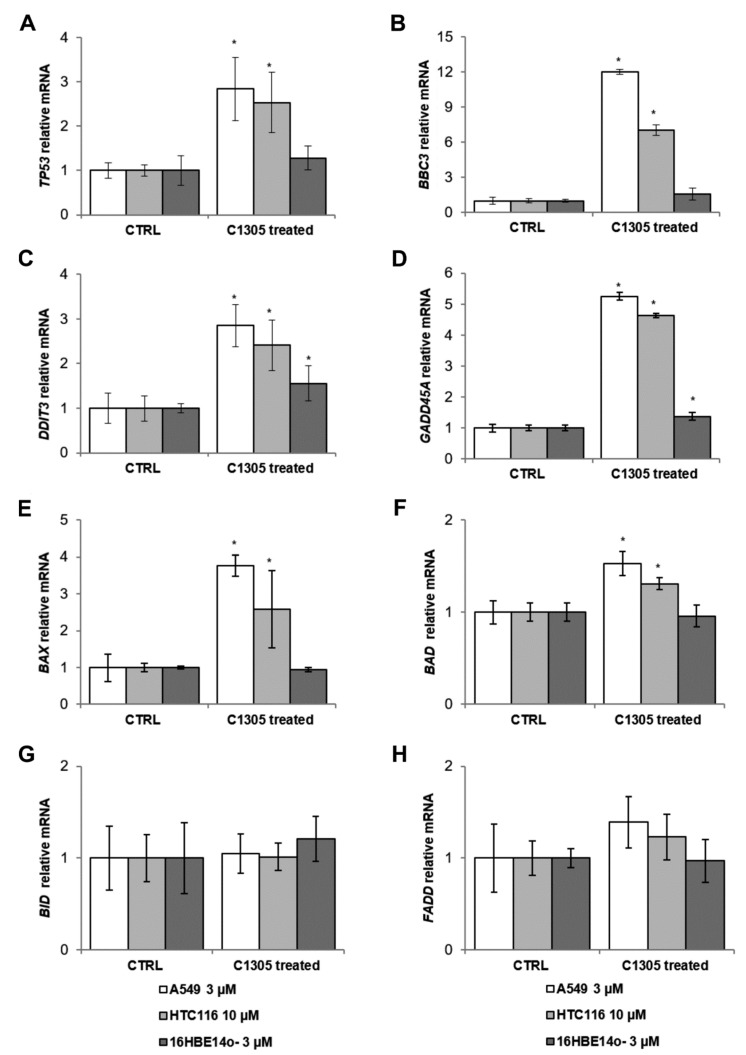
Exposure of A549 and HTC 116 cells to C-1305 activates transcriptional apoptotic signaling. A549 and 16HBE14o- cells were exposed to 3 µM C-1305 for 24 h, whereas HTC 116 cells were exposed to 10 µM C-1305 for 24h and levels of (**A**) TP53, (**B**) BBC3, (**C**) DDIT3, (**D**) GADD45A (**E**) BAX, (**F**) BAD, (**G**) BID, and (**H**) FADD were measured with qPCR. The results from three independent experiments (*n* = 6) are plotted normalized to TBP mRNA levels and expressed as a fold-change over the DMSO vehicle controls. Error bars represent standard deviations. Significant changes (*p* < 0.05) are marked with an asterisk.

**Figure 4 cancers-12-00864-f004:**
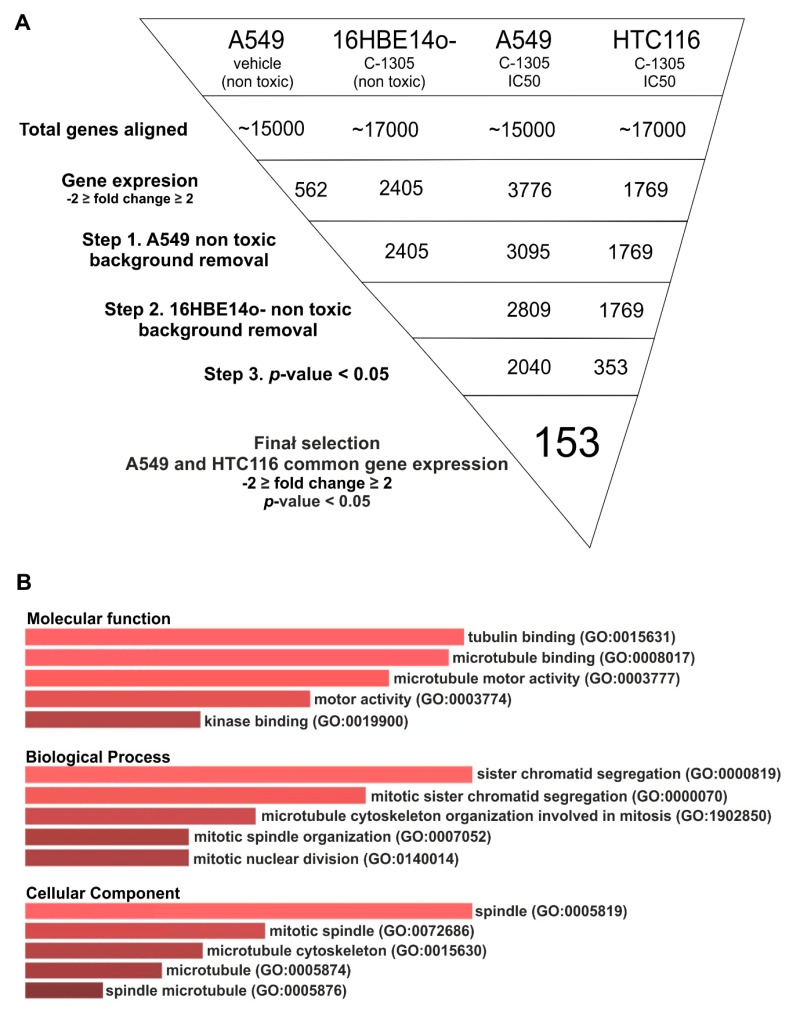
Schematic of analysis and data filtration and analysis performed to identify gene ontology (GO) terms of significantly dysregulated genes. (**A**) Schematic of analysis and data filtration Following the flirtation of gene transcript potentially related to A549 growth and no cytotoxic effects in 16HBE14o-cells. The common gene transcripts dysregulated in both A549 and HTC 116 were selected. Well-established gene transcripts (no predicted gene transcripts, gene names starting with “LOC”) with greater than 10 RPKMs per sample and with significance (*p* ≤ 0.05) greater or equal to a 2-fold change in expression between C-1305-treated and control groups were used in pathway analyses. (**B**) The GO clusters are depicted for selected genes; clusters are listed followed by the enrichment score calculated by Enricher, which is used to determine the percentage of the pie chart. The darker the color, the more enriched the cluster.

**Figure 5 cancers-12-00864-f005:**
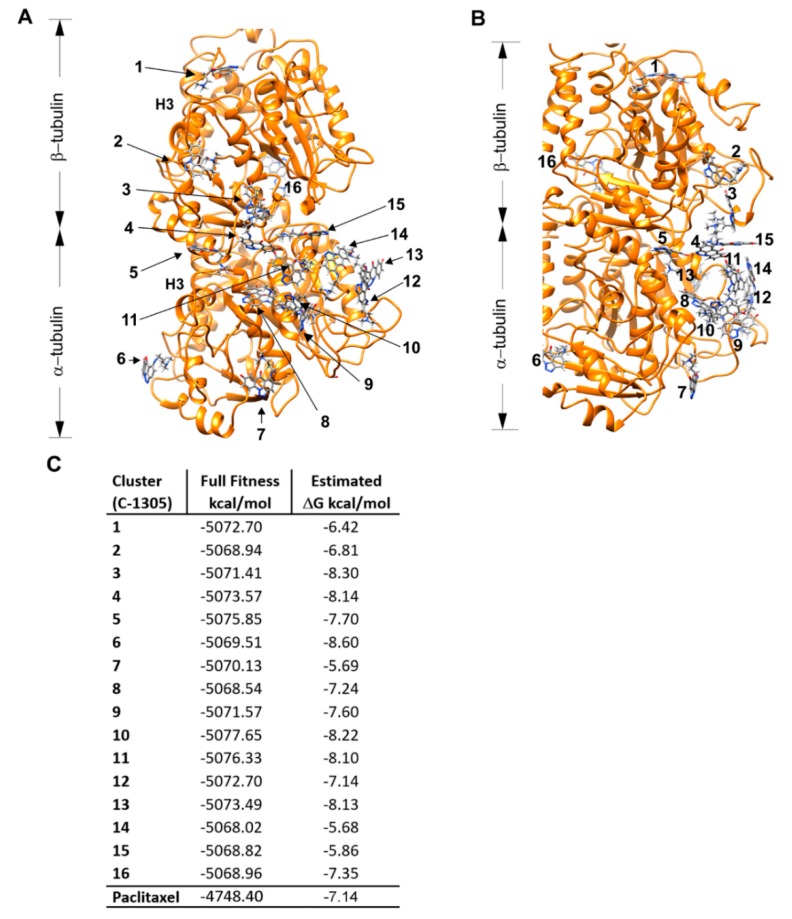
Molecular docking identifies potential C-1305 binding to the α/β tubulin dimer. (**A**) Predicted interactions between tubulin dimer and a C-1305 molecule. The favorable binding sites were identified by judging from the largest number of conformations in the same position with the lowest binding energy. The analysis showed minor differences among the top results of each binding site for C-1305, supporting the robustness of the calculations. Additionally, the superposition of the structure of the paclitaxel co-crystallized with the tubulin (PDB accession code 1TUB) onto that obtained from the docking of paclitaxel yielded a good fitting of both ligands. (**B**). A close-up of the representative snapshots of the simulations depicting the interactions of C-1305 at the luminal side of tubulin. (**C**) Classification of results obtained from the docking of C-1305 into tubulin by SwissDock. The Full Fitness and ΔG values (kcal/mol) for the most favorably ranked complex (cluster Rank = 0) are shown. Molecular graphics and analyses were performed with UCSF Chimera, developed by the Resource for Biocomputing, Visualization, and Informatics at the University of California, San Francisco [61]. Protein (1TUB) is shown as ribbon and ligands are shown as sticks, colored by element. The control structure of tubulin, with colchicine and molecular docking results for paclitaxel, are provided in Appendix A respectively. In Appendix A, we will add the three-dimensional structure of the α/β-tubulin dimer (PDB: 1TUB) [62] used for molecular docking, showing that paclitaxel binds to a unique pocket on β-tubulin near GDP binding site, which controls molecular docking of the paclitaxel molecule and tubulin protein and tubulin dimer crystal structure (1TUB) superimposed with the computed tubulin dimmer allosteric structure. The ZINC96006020 (Paclitaxel) entry obtained from the ZINC database (http://zinc.docking.org) was used for ligand structure generation [63].

**Figure 6 cancers-12-00864-f006:**
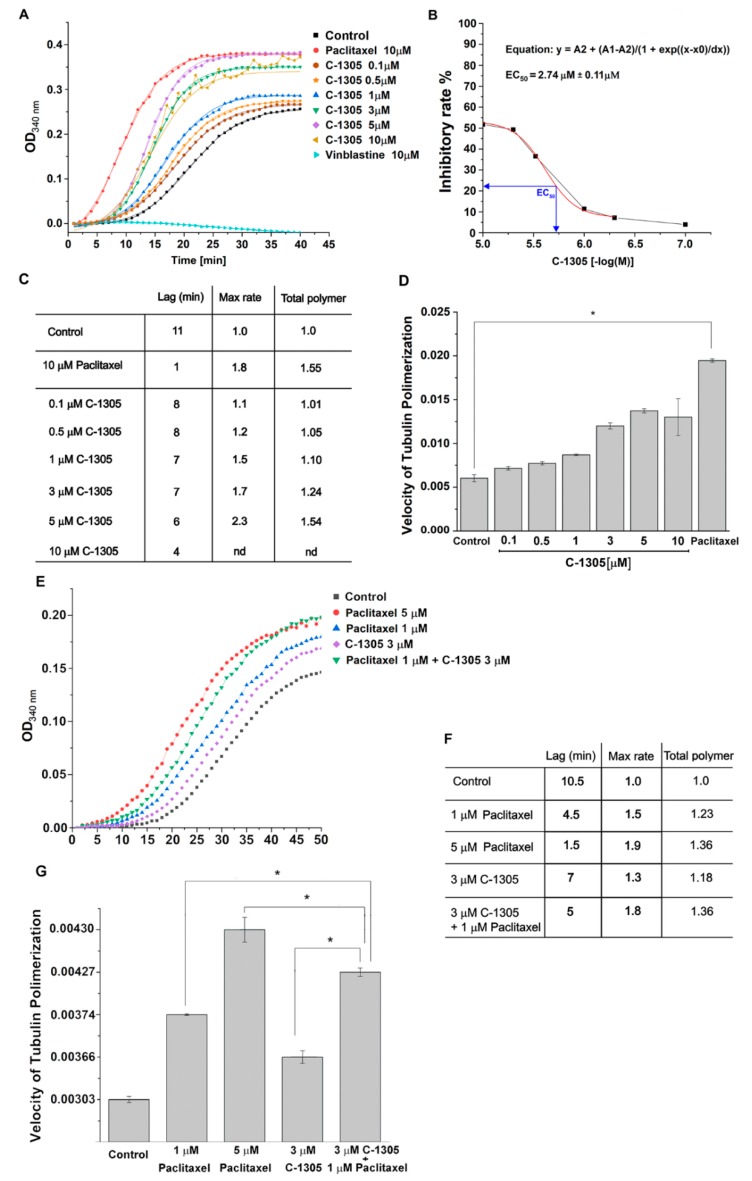
C-1305 acts as an efficient tubulin polymerization stabilizing agent in vitro and complements paclitaxel effects on tubulin polymerization (**A**) The kinetics of the polymerization of 3.0 mg/mL purified tubulin with microtubule stabilizers was monitored turbidimetrically at a 340 nm wavelength (OD_340 nm_). The increasing concentration of tubulin polymers as a function of time displays the characteristic sigmoidal curve for tubulin filament formation. The curve is typically divided into the lag phase (nucleation), the growth phase (polymerization), and the plateau phase [64]. The square data points in a line in each graph indicate microtubule polymerization alone (control). Vinblastine, a destabilizer of tubulin polymerization, was used as a control. C-1305 stabilized the polymerization of tubulin in a cell-free system in a dose-dependent manner. (**B**) The EC_50_ rate was calculated as the ratio of the OD_340 nm_ change obtained with C-1305 treatment to the OD_340 nm_ change of the control [65]. Due to observed tubulin oscillation at high doses of C-1305, to calculate EC_50_ for 10 µM C-1305 concentration, we used a mean value from three independent experiments. (**C**) The quantization of tubulin polymerization parameters was calculated as described in [66]. Lag indicates the time required to achieve exponential microtubule polymerization. The maximum relative rate and total polymerization have been normalized to non-drug-treated tubulin (control). The graph is representative of three independent experiments. (**D**) The velocity of tubulin polymerization in the growth phase after C-1305 treatment was calculated as described previously [67]. The polymerization velocity was calculated based on the ratio of the OD_340 nm_ change to the reaction time. The velocity in the absence of stabilizers was taken as 100%. * *p* < 0.05 versus control. Dose-dependent effect of C-1305 or paclitaxel on in-vitro microtubule formation. The level of polymerization was measured by an increase in turbidity (OD) at 340 nm. (**E**) The synergic/additive effects of concomitant C-1305 and paclitaxel treatment on tubulin polymerization. (**F**) The quantization of tubulin polymerization parameters was calculated as described in [66]. Data presented in the Appendix A graph was used for calculation. Lag indicates the time required to achieve exponential microtubule polymerization. The maximum relative rate and total polymerization have been normalized to non-drug-treated tubulin (control). The graph is representative of three independent experiments. (**G**) The velocity of tubulin polymerization in the growth phase after C-1305, paclitaxel or concomitant C-1305 and paclitaxel treatment. The velocity in the absence of stabilizers was taken as 100%. * *p* < 0.05 versus control.

**Figure 7 cancers-12-00864-f007:**
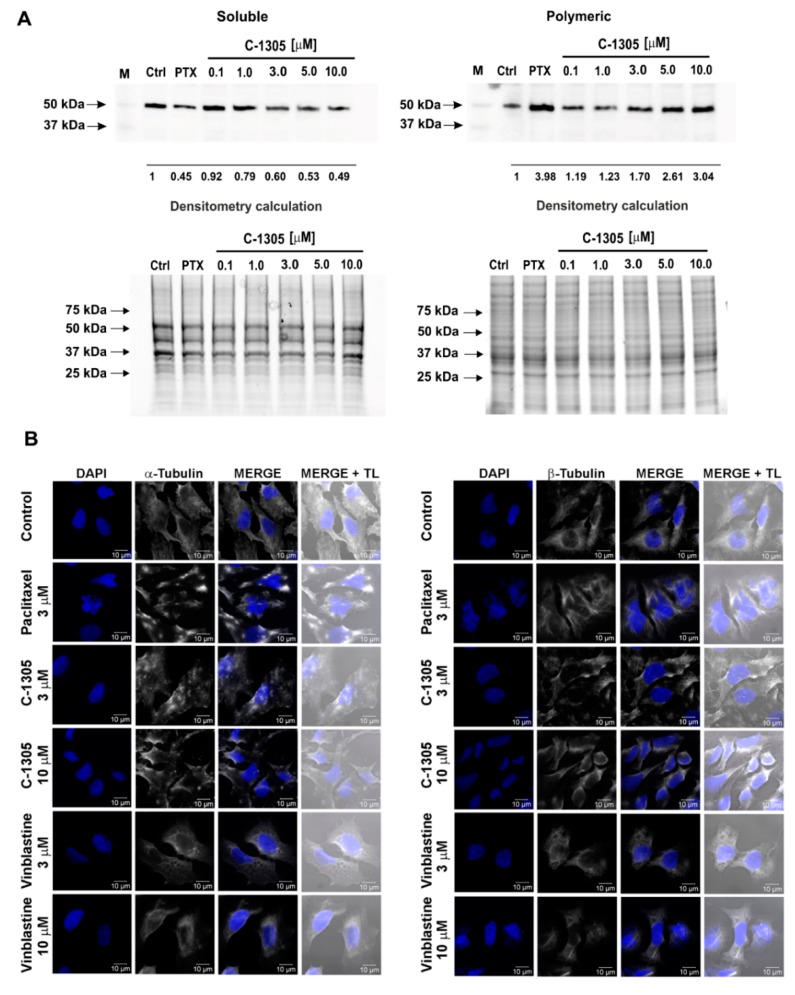
C-1305 stabilizes polymeric tubulin and changes the microtubule network in A549 cells. (**A**) SDS-PAGE (bottom panel) and Western blot analysis (top panel) of soluble and polymeric tubulin fractions in A549 cells after C-1305 treatment. Western blot analysis of soluble (S) and polymeric (P) tubulin fractions using rabbit primary antibody against β-tubulin (ab18207, Abcam) and HRP-conjugated secondary antibodies (Bio-Rad) and detected using ECL (Amresco). The molecular weight markers are indicated by the arrows on the left (Precision Plus Protein™ Kaleidoscope™ Prestained Protein Standards BioRad, #1610375). Densitometry was performed using Image Lab software v. 4.1 (Bio-Rad) and Gel-Pro Analyzer v. 4.0 (Media Cybernetics L.P); the amount in each band was calculated in relation to the corresponding band of the control lane. The bottom panel total protein loading controls SDS-PAGE of soluble (S) and polymeric (P) tubulin fraction isolated from A549 cells after 24h of treatment. The results of the soluble and polymeric tubulin fractions in A549 cells treated with C-1305 for 6 h are shown in Appendix A. Raw Western blot files are shown in Appendix A. (**B**) The C-1305 effects on the microtubule network were observed by immunofluorescence using a Confocal Scanning Laser Microscope (LSM) and Transmitted Light (TL). A549 cells were incubated in the absence and the presence of C-1305 (3 or 10 µM), paclitaxel (3 µM) and vinblastine (3 or 10 µM) for 24 h. Cells were fixed and processed for immunostaining with antibodies against α-tubulin (left panel) and β-tubulin (right panel) for 2 h, then reacted with Alexa fluoro-conjugated secondary antibody. The scale bar is 10 μm. Paclitaxel and vinblastine were used as the reference compound.

**Figure 8 cancers-12-00864-f008:**
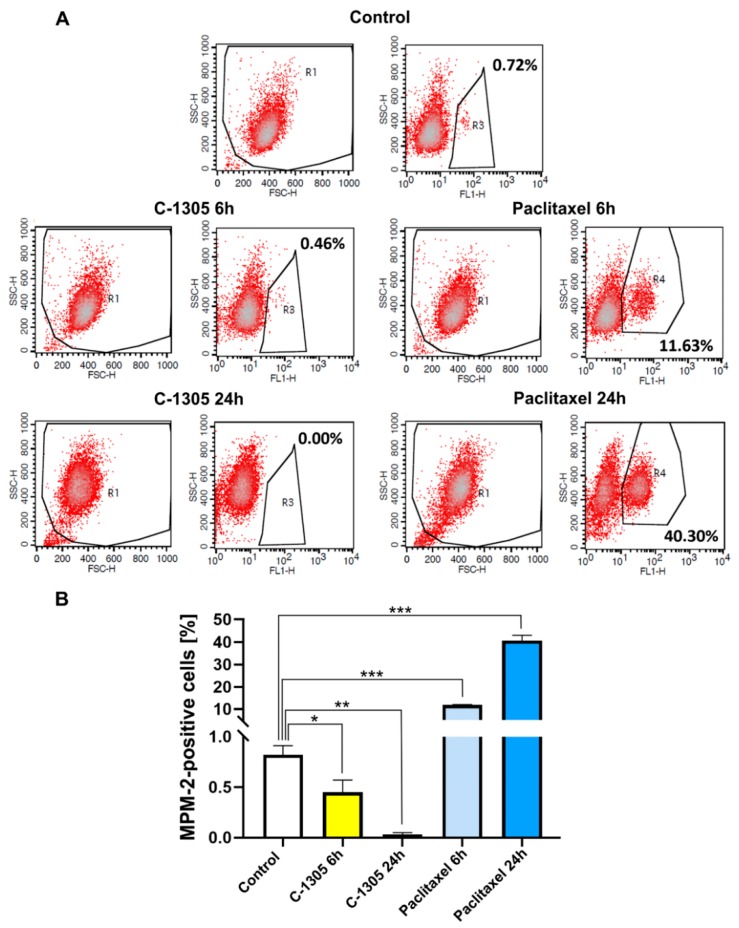
C-1305 induces G2 cell arrest in A549 cells. Cytometric analysis of MPM-2-positive cells after C-1305 or paclitaxel treatment (**A**). Representative dot blot of mitotic, MPM-2-positive cells in A549 cell line treated with C-1305 or paclitaxel for 6 or 24 h, respectively. MPM-2-positive cells were gated, quantified, and shown as a percentage next to the gate. (**B**). Percentage of MPM-2-positive cells presented as means ± SD from three independent experiments. * *p* < 0.05; ** *p* <0.001; *** *p* < 0.0001 versus control cells.

**Table 1 cancers-12-00864-t001:** Function of genes being significantly changed upon C-1305 treatment (*p* < 0.05) as assigned with the Gene Ontology databases and verified with the literature.

Gene	Function	Reference
*BBC3*	Mediator of p53/TP53-dependent/independent apoptosis	[20]
*DDIT3*	Multifunctional transcription factor in the response to cell stresses; induces cell cycle arrest and apoptosis in response to endoplasmic reticulum (ER) stress	[21,22]
*BAX*	Regulator of the mitochondrial apoptotic process	[23]
*BAD*	Promotes cell death	[23]
*GADD45A*	ER stress dependent/apoptotic	[24]
*BID*	Pro-apoptotic member of the B-cell lymphoma-2 (Bcl-2) protein family	[25]
*FADD*	Apoptotic adaptor molecule that recruits caspases to the activated Fas or TNFR-1 receptors.	[26]
*TP53*	Tumor suppressor in many tumor types; induces growth arrest or apoptosis	[27]
*AURKB*	Regulator of mitosis involved in the bipolar attachment of spindle microtubules to kinetochores	[28]
*PLK1*	Kinetochore assembly	[29]
*CDK1*	Regulate mitotic entry, centrosome separation and spindle assembly	[30]
*KIFC1*	Regulate chromosome congression and alignment and bipolar spindle formation	[31]
*KIF11*	Establishing a bipolar spindle during mitosis	[32]
*KIF14*	Regulate chromosome congression and alignment	[31]
*KIF15*	Involved in mitotic spindle assembly	[33]
*KIF18A*	Regulate chromosome congression and alignment	[34]
*KIF18B*	Microtubule movement and depolymerization	[35]
*KIF20A*	Cytokinesis	[31]
*KIF22*	Spindle formation and the movements of chromosomes during mitosis	[36]
*KIF23*	Cytokinesis	[31]
*KIF4A*	Anaphase spindle dynamics and cytokinesis	[31]
*CDCA8*	Mitosis regulation, microtubule stabilization and spindle assembly	[37]
*BIRC5*	Chromosome alignment and segregation during mitosis and cytokinesis	[38]
*CCNB1*	Control of the cell cycle at the G2/M (mitosis) transition	[39]
*CCNB2*	Control of the cell cycle at the G2/M (mitosis) transition	[40]
*CDC20*	Mitotic regulation of the human anaphase	[41]
*LMNB1*	Interact with chromatin and promote senescence	[42]
*CENPF*	Regulate kinetochore function and chromosome segregation in mitosis	[43]
*CENPJ*	Inhibits cell proliferation and induces apoptosis after G2/M arrest	[44]
*CENPK*	Assembly of kinetochore proteins, mitotic progression and chromosome segregation	[45]
*CENPN*	Assembly of kinetochore proteins, mitotic progression and chromosome segregation	[46]
*CENPQ*	Assembly of kinetochore proteins, mitotic progression and chromosome segregation	[47]
*FAM83D*	Cell proliferation, growth, and migration	[48]
*GAS2L3*	Cytokinesis/Stabilize the formation of the actin and microtubule network	[49]
*NUSAP1*	Promote the organization of mitotic spindle microtubules	[50]
*PRC1*	Cytokinesis and microtubule assembly	[51]
*SKA1*	Chromosome segregation and microtubule depolymerization	[52]
*STMN1*	Regulation of the microtubule filament system by destabilizing microtubules	[53]
*TP53I3*	Generation of reactive oxygen species (ROS)/Apoptosis	[54]
*TP53INP1*	Proapoptotic protein and regulator of transcription and autophagy	[55]
*SNAI2*	Transcriptional repressor	[56]
*PARP*	Mediates poly-ADP-ribosylation of proteins/DNA repair	[57]

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
