# Peer review of "Utilizing Genome-Wide mRNA Profiling to Identify the Cytotoxic Chemotherapeutic Mechanism of Triazoloacridone C-1305 as Direct Microtubule Stabilization"

_cancers, 2020, doi:10.3390/cancers12040864_

Round 1

Reviewer 1 Report

The work presented in this article titled "Utilizing genome-wide mRNA profiling to identify the cytotoxic chemotherapeutic mechanism of triazoloacridone C-1305 as direct microtubule stabilization" provides the mechanistic action of C-1305 specifically in lung cancer cells. The novelty of the study lies in understanding the mechanism of action rather than mRNA profiling which is now a common method of choice to determine changes in gene expression with respect to drugs. 

The introduction focusses on C-1305. The authors need to also emphasize and expand on the choice of cells in the study with respect to C-1305. 

Figure 2, what is the rationale for choosing varying lower concentrations in the conductance study of the drug with respect to A549, HCT116 and 16HBE14o-?  125 nM and 12.5 nM for A549, 250 nM and 125 nM for HCT116 and 25, 2.5 and 5 microM for 16HBE14o-

The mRNA profiling gene expression experiments were done using both A549 and HCT116. However, it is not clear why the later half of the study focused only on A549 cells. Figures 6, 7, and 8 are only on A549 . Please provide an explanation as to why only this cell line was chosen for the further mechanistic analysis of C-1305.

A major concern is the choice of controls for gene expression analysis. For A549 there was a non cancerous lung cell. However, for HCT116 the correction was made to the non cancerous lung cells.

The molecular docking study is interesting and provides new information regarding C-1305 and microtubule stabilization. Does C-1305 freely cross the plasma membrane in order for it to act on the microtubules?

Figure 7 the western blots and SDS page did not include the 3 microM concentration that was the IC50 for A549 cells.

All minor corrections are made directly in the pdf. Please go through it and make the necessary changes.

Reviewer 2 Report

In this manuscript, the authors measured the C-1305 cytotoxicity in some cell lines and used RNAseq to identify the change of gene expression. They reported C-1305 as a tubulin-binding agent that promotes tubulin polymerization and G2 cell cycle arrest. The study is interesting and will potentially contribute to the understanding of the mode of action of C-1305 in vitro. However, I have several questions.

The authors suggested, “Taken together, the results of the RNAseq-based approach suggested that the C-1305 cytotoxicity may result from the direct disorganization of the microtubule skeleton”. This is odd. Most of the genes in the list are related to cell cycle or apoptosis. Generally, disruption of any gene/pathway that promotes cell cycle progression will lead to a change like this. There’s no indication that the disorganization of the microtubule skeleton is to be attributed to. Then, the authors suggested, “The most straightforward potential mechanism explaining such an effect on the microtubule network would consist of C-1305’s direct interaction with tubulin.” This makes no sense. Why the mRNA change would result from C-1305’s direct interaction with tubulin? Again, disruption of any gene/pathway other than direct interaction with tubulin can lead to the gene expression change. What’s the real rational for those tubulin binding and stabilization assays?

The authors reported the C-1305’s effect on tubulin dynamics and cell cycle. While its effect on cell cycle is expected, its direct interaction with tubulin is new and interesting. But the cell-based experiments are not convincing yet. Because C-1305 treatment caused cell cycle arrest, and it’s known that cell cycle progresses with tubulin dynamics, it’s possible that the C-1305’s effect on tubulin is secondary. In another words, the authors were comparing the tubulin at two different cell cycle stages in the experiments in Figure 7.

Some minor points:

In the introduction section, line 83 to line 88 is a copy-paste from the journal instruction for authors.

Line 421, should “the selection of 143 transcripts” be 153 transcripts?

The authors need to have a careful proofreading.

Round 2

Reviewer 1 Report

NA

Reviewer 2 Report

The revised  manuscript addressed most of my comments and is now suitable for publication.